# LUNCH: Adaptive Balancing of Continual Learning via Hyperparameter Uncertainty

## Abstract

Continual learning (CL) is characterized by learning sequentially arriving tasks and behaving as if they were observed simultaneously. In order to prevent catastrophic forgetting of old tasks when learning new tasks, representative CL methods usually employ additional loss terms to balance their contributions (e.g., regularization and replay), modulated by deterministic hyperparameters. However, this strategy struggles to accommodate real-time changes in data distributions and is also lack of robustness to subsequent unseen tasks, especially in online scenarios where CL is performed with a one-pass data stream. Inspired by adaptive weighting in multi-task learning, we propose an innovative approach named Learning UNCertain Hyperparameters (LUNCH) for adaptive balancing of task contributions in CL. Specifically, we formulate each CL-relevant hyperparameter as a function of optimizable uncertainty under homoscedastic assumption and ensure its training stability through the exponential moving average of network parameters. We further devise an evaluation protocol that moderately adjusts the hyperparameter values and reports their impact on performance, so as to analyze the sensitivity of these sub-optimal values in realistic applications. We perform extensive experiments to demonstrate the effectiveness and robustness of our approach, which significantly improves online CL in a plug-in manner (e.g., up to 11.26% and 5.64% on Split CIFAR-100 and Split Mini-ImageNet, respectively) as well as offline CL.[1]

## 1 Introduction

The ability of continual learning (CL) is critical for artificial intelligence systems to accommodate real-world changes, yet limited by catastrophic forgetting of old tasks when learning new tasks (Wang et al., 2024a; McClelland et al., 1995). In order to strike an appropriate balance between task contributions within the same parameter space, representative CL methods often employ additional loss terms to preserve previously learned knowledge, such as regularization of parameter changes (Kirkpatrick et al., 2017; Zenke et al., 2017) and replay of a few old training samples (Buzzega et al., 2020; Rebuffi et al., 2017). In general, the strength of these loss terms is regulated by deterministic hyperparameters obtained from a grid search (Chaudhry et al., 2018; Wang et al., 2023a). However, this strategy is sub-optimal in performance as it struggles to adapt to real-time changes of data distributions within the observed task sequence, and is also lacking robustness to subsequent unseen tasks. These critical challenges tend to be more significant in online CL where each task is learned from a one-pass data stream (Fini et al., 2020; Zhang et al., 2022).

In this regard, we analyze in depth the role of CL-relevant hyperparameters in balancing task contributions. We first formulate representative CL methods with a shared mathematical form of the loss function. Besides a loss term for learning the current task, the loss function typically involves additional loss terms that preserve previously learned knowledge in terms of the parameter space and the output space with corresponding hyperparameters. These loss terms amount to approximate multi-task learning (MTL) for all tasks ever seen, i.e., the upper bound of CL, while largely avoiding the use of old training samples. In MTL, adaptive weighting of task contributions in hyperparameters has been shown to be an effective strategy compared to fixed weighting (i.e., using deterministic hyperparameters) (Kendall et al., 2018; Liu et al., 2019; 2022; Lin et al., 2021), but

---

[1]Our code is included in Supplementary Materials for examination and will be released upon acceptance.

remains under-explored and highly non-trivial for CL due to the dynamic and unpredictable nature of data distribution.

Based on the above analysis, we present Learnable UNCertain Hyperparameters (LUNCH), an innovative approach that enables adaptive balancing of task contributions in CL. Specifically, we formulate each CL-relevant hyperparameter as a function of optimizable uncertainty, which is initialized high and then decreases during the learning of changes in data distributions. Under the homoscedastic uncertainty assumption, we derive probabilistic implementations for the loss terms of the parameter space and output space, corresponding to regression and classification problems, respectively. Whenever a new task is introduced, the uncertain hyperparameters need to be refreshed to re-balance the contributions, resulting in a performance degradation known as the "stability gap" (De Lange et al., 2022). In this regard, we perform exponential moving average of network parameters along the training trajectory, so as to stabilize training upon reinitialization.

We perform extensive experiments to evaluate our approach. Beyond the widely-used average accuracy for overall performance, we consider two additional evaluation metrics including the average anytime accuracy and the worst-case accuracy for real-time changes in data distributions. We further evaluate the sensitivity of sub-optimal hyperparameter values through analyzing their impact under moderate adjustments. Our approach demonstrates outstanding performance with significant improvements in effectiveness and robustness across various *online* CL benchmarks, benefiting recent strong baselines in a plug-in manner (e.g., up to 11.26% and 5.64% on Split CIFAR-100 and Split Mini-ImageNet, respectively) and also remarkably facilitate *offline* CL.

Our contributions can be summarized as follows: (1) We perform an in-depth analysis of CL-relevant hyperparameters under a unified framework of representative CL methods and task balancing strategies in MTL; (2) We propose an innovative approach that incorporates optimizable uncertainty into CL-relevant hyperparameters for adaptive balancing of task contributions, coupled with exponential moving average of network parameters to address the stability gap; and (3) Our approach significantly improves the effectiveness and robustness of CL, validated by extensive experiments.

## 2 RELATED WORK

**Continual Learning (CL)**, also known as incremental learning or lifelong learning, aims to overcome catastrophic forgetting of old tasks when learning new tasks (Wang et al., 2024a; McClelland et al., 1995). Numerous efforts have been devoted into addressing this challenging issue. A majority of representative methods attempt to strike an appropriate balance between task contributions within the same parameter space. For example, regularization-based methods employ explicit regularization terms to stabilize network parameters and simulate behaviors of the old model (Kirkpatrick et al., 2017; Buzzega et al., 2020; Li & Hoiem, 2017). Meanwhile, replay-based methods approximate and recover the old data distributions through preserving a small memory buffer or learning a generative model (Buzzega et al., 2020; Shin et al., 2017; Aljundi et al., 2019). Other methods that optimize network parameters in different parameter spaces are often collectively referred to as architecture-based methods (Serra et al., 2018; Kang et al., 2022; Rusu et al., 2016), which explicitly avoid the problem of balancing task contributions. However, this kind of method typically requires the oracle of task identity at test time in order to select an appropriate parameter space, and is therefore not prioritized in this work. Based on the availability of training samples, the widely-used CL setups can be categorized into online CL and offline CL (detailed in Section 3.1), with the former being considered realistic yet much more challenging.

**Hyperparameter of CL**. For representative CL methods, an appropriate management of hyperparameters (e.g., learning rates, regularization strengths, memory buffer sizes, etc.) is critical for achieving outstanding performance across tasks. A primary consideration is task balancing, which ensures that the model maintains performance on old tasks while learning new ones (Cha & Cho, 2024; Yildirim et al., 2023). As the upper bound of CL, multi-task learning (MTL) attempts to address this problem through various weighting strategies: fixed weighting assigns constant importance to each task, while adaptive weighting adjusts the importance based on task difficulty or model performance (Kendall et al., 2018; Liu et al., 2019; 2022; Lin et al., 2021). Despite the effectiveness in MTL, adaptive weighting of CL-relevant hyperparameters is remarkably challenging due to the dynamic and unpredictable nature of data distribution, and therefore remains largely under-explored in literature.

## 3 PRELIMINARIES

In this section, we first describe the problem formulation of CL with a unified framework of representative methods. We then analyze the selection of hyperparameter(s) in CL on the basis of the inherent connections between CL and MTL.

### 3.1 PROBLEM FORMULATION

Let us consider a sequence of tasks defined by a collection of respective training sets $S = \{\mathcal{D}_1, \cdots, \mathcal{D}_T\}$, where $\mathcal{D}_t$ for each task $t$ consists of several data-label pairs $(x_t, y_t)$ with $x_t \in \mathcal{X}_t$ and $y_t \in \mathcal{Y}_t$. The goal of CL is to learn a mapping $f_\theta : \bigcup_{t=1}^{T} \mathcal{X}_t \to \bigcup_{t=1}^{T} \mathcal{Y}_t$ parameterized by trainable parameters $\theta$ with sequentially arriving $\mathcal{D}_t$, so as to achieve superior performance on all observed tasks (Wang et al., 2024a). Since previous training samples are often unavailable at the current training stage, it remains extremely challenging to strike an appropriate balance between old and new tasks, resulting in catastrophic forgetting (i.e., $f_\theta$ abruptly and dramatically forget previously learned knowledge upon new information). Regarding specific setups, training samples for each task can be reused for multiple epochs in *offline* CL, but arrive as a one-pass data stream in *online* CL, which greatly adds to the challenge.

To alleviate catastrophic forgetting when optimizing $\theta$ within the *same* parameter space, many representative methods have been proposed for CL. These methods can be classified into regularization-based methods, which incorporate additional regularization term(s) to stabilize knowledge of the parameter space and the output space (Kirkpatrick et al., 2017; Li & Hoiem, 2017), as well as replay-based methods (Aljundi et al., 2019; Zhang et al., 2022), which preserve some old training samples with a small memory buffer. In particular, replay is often coupled with regularization (Buzzega et al., 2020; Rebuffi et al., 2017) to encourage the current model $f_\theta$ to mimic the behaviors of the old model $f_{\theta^*}$ with parameters $\theta^*$ when processing old training samples.

Following a recent work (Wang et al., 2024b), these two kinds of methods can be described as shared mathematical forms under a unified framework:

$$\mathcal{L}_{\text{CL}} = \lambda_n \underbrace{\mathcal{L}_n(x, y)}_{\text{new task}} + \lambda_o \underbrace{\mathcal{L}_o\left(f_\theta(x), z\right)}_{\text{output space}} + \lambda_p \underbrace{\mathcal{L}_p\left(\theta, \theta^*\right)}_{\text{parameter space}}, \tag{1}$$

where $\mathcal{L}_n$ denotes the loss function for learning each new task. $\mathcal{L}_o$ and $\mathcal{L}_p$ restrict update rates in output space and parameter space, respectively. The example definitions of $\mathcal{L}_o$ and $\mathcal{L}_p$ will be described latter in Table 1. It can be seen that the contributions of new and old tasks are explicitly regulated by the hyperparameters $\{\lambda_n, \lambda_o, \lambda_p\}$.

### 3.2 HYPERPARAMETERS IN CONTINUAL LEARNING

With Eq. (1), we further analyze the selection of hyperparameters in representative CL methods. Similar to regular machine learning methods, the optimal hyperparameter values for CL are usually obtained by repeated iterations of the task sequence $S = \{\mathcal{D}_1, \cdots, \mathcal{D}_T\}$, which can be further divided into two strategies. One is to run the first several tasks iteratively to determine the hyperparameter values and use them to learn subsequent tasks (Chaudhry et al., 2018; Pham et al., 2021); the other is to run the entire task sequence iteratively in different orders to determine the hyperparameter values and provide a sensitivity analysis of them (Yildirim et al., 2023; Cha & Cho, 2024; Van de Ven et al., 2022). Both strategies assume relatively stable changes in data distribution and employ deterministic hyperparameters for CL, making it difficult to adapt to real-world scenarios that are highly dynamic and unpredictable (Semola et al., 2024; Cha & Cho, 2024). In addition, manually adjusting these hyperparameters over time is both costly and impractical.

In fact, MTL is often considered to be the upper bound of CL. Both CL and MTL aim to achieve the same objective, i.e., to find a solution $\theta$ that performs well for all observed tasks, with the main difference being whether $S$ is provided sequentially or simultaneously. Formally, the objective of MTL can be defined as follows:

$$\mathcal{L}_{\text{MTL}} = \sum_i w_i \mathcal{L}_i, \tag{2}$$

Table 1: Definition of representative CL methods that target the same parameter space. $F$ is the Fisher information matrix to approximate the importance of the network parameters. $M$ denotes the memory buffer consisting of a few old training samples. $x_{\text{aug}}$ is the augmentation of $x$. $z$ is the output logit of the old models in the training trajectory.

| Method | Regularization Loss | Replay Loss |
|---|---|---|
| EWC (Kirkpatrick et al., 2017) | $(\theta - \theta^*)^\top F(\theta - \theta^*)$ | - |
| ER (Shin et al., 2017) | - | $\mathbb{E}_{(x,y) \in M}(\mathcal{L}(x,y))$ |
| MIR (Aljundi et al., 2019) | - | $\max \mathbb{E}_{(x,y) \in M}(\mathcal{L}(x,y))$ |
| DER (Buzzega et al., 2020) | $\mathbb{E}_{(x,y) \in M}(\|f_\theta(x) - z\|_2^2)$ | - |
| DER++ (Boschini et al., 2022) | $\mathbb{E}_{(x,y) \in M}(\|f_\theta(x) - z\|_2^2)$ | $\mathbb{E}_{(x,y) \in M}(\mathcal{L}(x,y))$ |
| RAR (Zhang et al., 2022) | - | $\mathbb{E}_{(x,x_{\text{aug}},y) \in M}(\mathcal{L}(x_{\text{aug}},y))$ |

where $\mathcal{L}_i$ corresponds to the loss function for learning each task, and $w_i$ denotes the hyperparameter that regulates the task weight. The loss function of CL in Eq. (1) can be seen as an approximation of Eq. (2), with the use of old training samples largely avoided.

Although many MTL methods also employ deterministic hyperparameters selected from a grid search, even simplified into an unweighted form $w_i = w_j$, adaptive balancing of task contributions has proven to be a superior strategy (Kendall et al., 2018; Liu et al., 2019; 2022). In particular, the corresponding $\{w_i\}$ can be modeled as an optimizable function related to the relative confidence between tasks (Liu et al., 2019; Kendall & Gal, 2017). However, these adaptive weighting strategies remain under-explored and highly non-trivial for CL, due to the dynamic and unpredictable properties of data distributions. To this end, we aim to provide an innovative approach to address the above challenges, as detailed below.

## 4 LEARNABLE UNCERTAIN HYPERPARAMETERS (LUNCH)

In this section, we design an innovative adaptive weighting strategy for CL that optimizes CL-relevant hyperparameters according to training progress. We incorporate optimizable uncertainty into CL-relevant hyperparameters under the unified framework of representative CL methods, and further rectify the "stability gap" introduced by uncertainty refresh.

We first define the specific forms of several representative CL methods with Eq. (1), as shown in Table 1. These methods mainly focus on addressing the problem of catastrophic forgetting by limiting model updates in either parameter space or output space, so as to preserve previously learned knowledge. For adaptive balancing of task contributions, we propose to incorporate optimizeable parameters $\sigma^2$ (called "uncertainty") into the hyperparameter set $\{\lambda_n, \lambda_o, \lambda_p\}$ in Eq. (1). From a Bayesian perspective, such uncertainty can capture the model's confidence in different types of tasks and accordingly adjust the CL process to balance new and old tasks (Guo et al., 2011). During the learning of each task in CL, the confidence in the predictive distribution $p(y|x,\theta)$ should gradually increase, and corresponding uncertainty $\sigma^2$ should gradually decrease from large to small and eventually stabilize at a certain value (Kendall & Gal, 2017).

In particular, we observe that the loss terms $\mathcal{L}_n$, $\mathcal{L}_o$ and $\mathcal{L}_p$ correspond to addressing a regression or classification problem (Bishop & Nasrabadi, 2006). Specifically, the *classification* problem predicts discrete labels (e.g., DER employs a cross-entropy loss $\mathcal{L}_o$ with a small memory buffer), while the *regression* problem predicts continuous numerical value (e.g., EWC employs a weighted squared loss $\mathcal{L}_p$ to stabilize parameter changes). Here we focus on homoscedastic aleatoric uncertainty (Kendall & Gal, 2017), whose corresponding hyperparameters $\{\lambda_n, \lambda_o, \lambda_p\}$ do not depend on specific input data, rather stay constant for all inputs and vary only between different tasks.

For *regression* problems, the predictive distribution can be defined as a Gaussian distribution under the Laplace approximation (Bishop & Nasrabadi, 2006), finding a Gaussian approximation to a continuous probability density:

$$p\left(y \mid f_\theta(x)\right) = \mathcal{N}\left(f_\theta(x), \sigma^2\right), \tag{3}$$

where $\sigma^2$ denotes the homoscedastic aleatoric uncertainty. Due to the space limit, the detailed derivation can be found in Appendix A.1. When using the mean squared error as the loss function

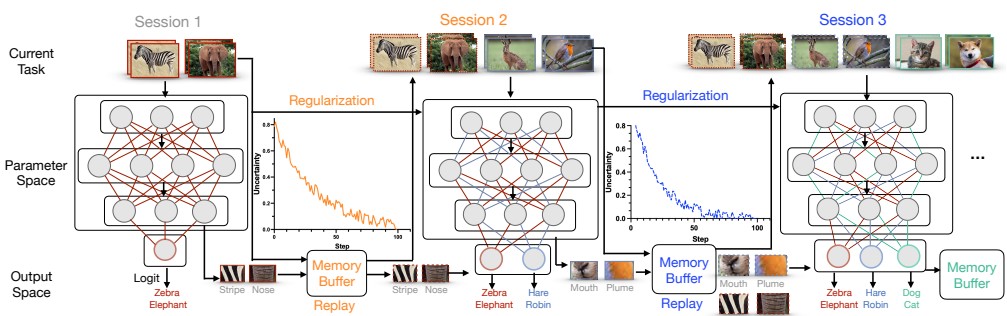

Figure 1: Demonstration of our approach. Session 2 and Session 3 shows that the proposed LUNCH incorporates optimizable uncertainty into CL-relevant hyperparameters, enabling adaptive balancing of task contributions in both parameter space and output space.

$\mathcal{L}$ for regression problem, the corresponding log-likelihood becomes:

$$-\log p\left(y \mid f_\theta(x)\right) \propto \frac{1}{2\sigma^2}\left\|y - f_\theta(x)\right\|^2 + \frac{1}{2}\log\sigma^2. \tag{4}$$

For *classification* problems, we use a Gibbs distribution to capture the predictive distribution scaled by the learnable "temperature" $\sigma^2$, which determines the flatness of discrete distribution (i.e., entropy) (Bishop & Nasrabadi, 2006):

$$p\left(y \mid f_\theta(x)\right) = \mathrm{Softmax}\left(\frac{1}{\sigma^2}f_\theta(x)\right), \tag{5}$$

where the $\sigma^2$ determines the degree of entropy divergence in the deterministic discrete distribution. Then, the corresponding log-likelihood becomes:

$$\log p\left(y = c \mid f_\theta(x)\right) = \frac{1}{\sigma^2}f_\theta(x) - \log\left[\sum_{c'=1}^{c}\exp\left(\frac{1}{\sigma^2}f_\theta^{c'}(x)\right)\right], \tag{6}$$

where $f_\theta^{c'}$ denotes the $c'$-th logit. To simplify the derivation in CL regarding its dynamic and unpredictable nature, the explicit trick $\frac{1}{\sigma^2}\sum_{c'=1}^{c}\exp\left(\frac{1}{\sigma^2}f_\theta^{c'}(x)\right) \approx \left(\sum_{c'=1}^{c}\exp\left(\frac{1}{\sigma^2}f_\theta^{c'}(x)\right)\right)^{\frac{1}{\sigma^2}}$ becomes an equality when $\sigma^2 \to 1$, which has been verified in empirically improving the performance. Due to the space limit, the derivation is detailed in Appendix A.2. Then, the overall log-likelihood of the predictive distribution is:

$$-\log p\left(y \mid f_\theta(x)\right) \approx \frac{1}{\sigma^2}\log\mathrm{Softmax}\left(y, f_\theta(x)\right) + \log\sigma^2. \tag{7}$$

Then, we can reformulate each CL-relevant hyperparameter in Eq. (1) as a function $\lambda(\sigma^2)$ of homoscedastic aleatoric uncertainty $\sigma^2$ based on the training progress. The overall objective function $\log p\left(y \mid x; \sigma_n^2, \sigma_o^2, \sigma_p^2\right)$ is defined as follows:

$$\log\left(p \mid x, y; \sigma_n^2, \sigma_o^2, \sigma_p^2\right) = \exp\left(-\log\sigma_n^2\right)\underbrace{\mathcal{L}_n(x,y)}_{\text{new task}} + \exp\left(-\log\sigma_o^2\right)\underbrace{\mathcal{L}_o\left(h_\theta(x), z\right)}_{\text{output space}}$$
$$+ \exp\left(-\log\sigma_p^2\right)\underbrace{\mathcal{L}_p\left(\theta, \theta^*\right)}_{\text{parameter space}} + \log\sigma_n^2 + \log\sigma_o^2 + \log\sigma_p^2, \tag{8}$$

where $\mathcal{L}_n(x,y)$, $\mathcal{L}_o(x,y)$ and $\mathcal{L}_p(x,y)$ represent the loss terms for CL. $\{\lambda_n(\sigma_n^2) = \exp\left(-\log\sigma_n^2\right), \lambda_o(\sigma_o^2) = \exp\left(-\log\sigma_o^2\right), \lambda_p(\sigma_p^2) = \exp\left(-\log\sigma_p^2\right)\}$ represent aforementioned uncertain hyperparameters. As uncertainty $\sigma^2$ increases, the relative weights of the loss function $\mathcal{L}$ decrease, and vice versa. The additional term $\log\sigma^2$ discourages rapid changes in $\sigma^2$, thus stabilizing the norm of relative weights. We also use the exponential mapping trick (Murphy, 2012) to

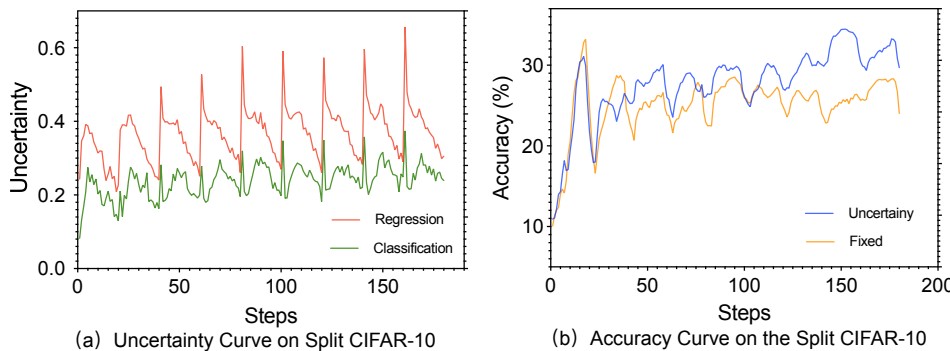

(a) Uncertainty Curve on Split CIFAR-10     (b) Accuracy Curve on the Split CIFAR-10

Figure 2: CL suffers substantial forgetting as each new task is introduced, followed by a phase of performance recovery. The experiment is performed with RAR on Split CIFAR-10.

ensure numerical stability, letting $f_\theta$ directly predict the log variance $\sigma^2$ to avoid division by zero. As shown in Fig. 2 (left), when they gradually learn representative features in a new task, the corresponding uncertainty about the predictive distribution gradually decreases and ultimately stabilizes at a fixed local optima (Kendall & Gal, 2017).

With significant changes in data distribution over the course of training (e.g., switching from $\mathcal{D}_{t-1}$ to $\mathcal{D}_t$), we need to refresh the uncertain hyperparameters for adaptation (i.e., the uncertainty is re-raised, and then gradually decreased):

$$\lambda_n\left(\sigma_n^2\right) = \lambda_n\left(\sigma_{n,\text{init}}^2\right); \lambda_o\left(\sigma_o^2\right) = \lambda_o\left(\sigma_{o,\text{init}}^2\right); \lambda_p\left(\sigma_p^2\right) = \lambda_p\left(\sigma_{p,\text{init}}^2\right). \tag{9}$$

This is accompanied by a critical challenge called the "stability gap" (De Lange et al., 2022): $f_\theta$ suffers substantial forgetting when starting to learn new tasks, followed by a phase of performance recovery (see Fig. 2, right).

To alleviate the mismatch between the refreshed hyperparameters and the current training progress, we employ the temporal ensemble (Laine & Aila, 2016) strategy to offset the bias. Specifically, we collect models along the training trajectory with exponential moving average (EMA), denoted as $f_{\theta^{\text{ema}}}$. For the current model $f_\theta$ parameterized by trainable $\theta$, the EMA at step $t$ is defined as:

$$\theta_t^{\text{ema}} = \frac{\beta\theta_{t-1}^{\text{ema}} + (1-\beta)\theta_t}{1-\beta^t} = \beta^t\theta_0^{\text{ema}} + \sum_{i=1}^t (1-\beta)\beta^{t-i}\theta_i, \tag{10}$$

where $\beta$ controls the strength of EMA, $\beta^t$ denotes $\beta$ raised to the power of $t$, and the parameter $\theta_i$ is at the $i$-th step in the training trajectory before $t$ step. The temporal ensemble of different models can enhance the stability of online CL, which has also been observed in a recent study (Soutif-Cormerais et al., 2023). To stabilize training when refreshing hyperparameters, the EMA strategy implicitly integrates hyperparameters from different training stages of the old tasks, providing an adaptive benefit for implementing the optimizable uncertainty.

Finally, we describe the LUNCH training procedure with a pseudo-code in Algorithm 1. The gray area highlights the key procedure. As can be seen, our approach is easy to implement and compatible with many representative CL methods, such as ER (Rolnick et al., 2019), DER (Buzzega et al., 2020), RAR (Zhang et al., 2022), etc.

## 5 EXPERIMENTS

In this section, we briefly describe the experimental setups and then analyze the experimental results.

### 5.1 EXPERIMENTAL SETUPS

**Benchmark**. Here we consider several benchmarks that are commonly used in the literature of online CL (Wang et al., 2023b). Specifically, CIFAR-10 dataset includes 10 classes of images sized $32 \times 32$, randomly divided into 5 disjoint tasks with 2 classes each. CIFAR-100 dataset includes

---

**Algorithm 1** Learnable UNCertain Hyperparameters (LUNCH) for CL

---

1: **Input**: Deep model $f_\theta$ with parameters $\theta$; uncertainty $\{\sigma_n^2, \sigma_o^2, \sigma_p^2\}$; datasets $\mathcal{S} = \{\mathcal{D}_1, \cdots, \mathcal{D}_T\}$.
2: **Initialization**: $\theta_0^{\text{ema}} = \theta_{\text{init}}$; $E = 1$ for online CL; $E > 1$ for offline CL.
3: **for** task $t = 1, \cdots, T$ **do**
4:     **if** $t = 1$ **then**
5:         **for** epoch $e = 1, \cdots, E$ **do**
6:             Update the parameters $\theta$ of $f_\theta$ only with loss $\mathcal{L}_n$
7:         **end for**
8:         Save the parameters $\theta$ of $f_\theta$ as $\theta_1^{\text{ema}}$
9:     **else**
10:         **for** epoch $e = 1, \cdots, E$ **do**
11:             Calculate the unified CL loss by Eq. (8).
12:             Update the parameters $\theta$ of $f_\theta$ and the uncertainty $\{\sigma_n^2, \sigma_o^2, \sigma_p^2\}$
13:             Update $\theta_t^{\text{ema}}$ by Eq. (10)
14:         **end for**
15:         Refresh the uncertainty $\{\sigma_n^2, \sigma_o^2, \sigma_p^2\}$ in Eq. (9)
16:     **end if**
17: **end for**

---

100 classes of images sized $32 \times 32$, randomly divided into 20 disjoint tasks with 5 classes each. Mini-ImageNet dataset includes 100 classes of images sized $84 \times 84$, randomly split into 20 disjoint tasks with 5 classes each. The common image resolution for the ImageNet-R dataset is $224 \times 224$, randomly divided into 20 disjoint tasks with 10 classes each (Hendrycks et al., 2021).

**Implementation**. We use Empirical Replay (ER) (Rolnick et al., 2019), Dark Experience Replay (DER) (Buzzega et al., 2020) and Repeated Augmented Rehearsal (RAR) (Zhang et al., 2022) as the main baselines for CL. Following the implementation of RAR (Zhang et al., 2022), we train a ResNet-32 backbone with an SGD optimizer of learning rate 0.1 in all experiments. The batch size is set to 32 for Split CIFAR-10/100, and 64 for both Split Mini-ImageNet and Split ImageNet-R. The memory buffer is unified to maintain 2000 training samples in total. For offline CL, the number of epochs is set to 100 that is sufficient for convergence on each task.

**Evaluation Metric**. We consider multiple evaluation metrics to provide a comprehensive analysis of CL. We first define the Final Average Accuracy (FAA) to effectively evaluate the overall performance after learning the last task $\mathcal{D}_T$. Formally, $\mathcal{D}_i$ represents the $i$-th task, while $f_{\theta_i}$ denotes the model parameterized by trainable $\theta_i$ at the $i$-th task:

$$\text{FAA} = \frac{1}{T} \sum_{i=1}^{T} A(\mathcal{D}_i, f_{\theta_T}). \tag{11}$$

However, FAA only provides a snapshot of the state after learning all tasks, ignoring performance during task transitions. To provide a more comprehensive assessment, we therefore consider two other metrics. One is the Average Anytime Accuracy (AAA), which measures the average performance on all observed tasks (De Lange et al., 2022).

$$\text{AAA} = \frac{1}{T} \sum_{j=1}^{T} \frac{1}{j} \sum_{i=1}^{j} A\left(\mathcal{D}_i, f_{\theta_j}\right). \tag{12}$$

Another is the Worst-Case Accuracy (WCA), which evaluates the performance of CL methods in the worst-case scenarios (De Lange et al., 2022). WCA first obtains the average minimum accuracy called minAcc in previous tasks and then combines it with the performance of current task (i.e., the

Table 2: Overall performance of *online* CL. The results of all methods are averaged over five runs with different random seeds and task orders.

| Benchmark | Method | $\frac{1}{4}\lambda$ | | | $\lambda$ | | | $4\lambda$ | | |
|---|---|---|---|---|---|---|---|---|---|---|
| | | FAA($\uparrow$) | WCA($\uparrow$) | AAA($\uparrow$) | FAA($\uparrow$) | WCA($\uparrow$) | AAA($\uparrow$) | FAA ($\uparrow$) | WCA ($\uparrow$) | AAA ($\uparrow$) |
| Split CIFAR-10 | ER | $43.51_{\pm0.40}$ | $38.58_{\pm5.01}$ | $57.12_{\pm4.06}$ | $34.77_{\pm8.01}$ | $32.97_{\pm6.47}$ | $54.98_{\pm2.67}$ | $34.78_{\pm3.22}$ | $32.97_{\pm2.23}$ | $43.26_{\pm3.24}$ |
| | w/ Ours | $47.50_{\pm0.58}$ | $40.59_{\pm4.96}$ | $59.73_{\pm3.38}$ | $45.64_{\pm1.14}$ | $39.24_{\pm3.04}$ | $58.22_{\pm3.68}$ | $47.64_{\pm3.99}$ | $39.24_{\pm2.87}$ | $53.07_{\pm3.65}$ |
| | $\Delta(\uparrow)$ | 3.99 | 2.01 | 2.61 | 10.87 | 6.27 | 3.24 | 12.86 | 6.27 | 9.81 |
| | DER | $36.55_{\pm0.22}$ | $36.05_{\pm0.22}$ | $35.69_{\pm0.23}$ | $47.63_{\pm4.81}$ | $41.46_{\pm8.15}$ | $56.57_{\pm0.84}$ | $15.92_{\pm0.84}$ | $15.91_{\pm0.84}$ | $30.92_{\pm0.54}$ |
| | w/ Ours | $39.02_{\pm1.77}$ | $38.97_{\pm1.83}$ | $42.70_{\pm3.36}$ | $54.98_{\pm3.55}$ | $45.78_{\pm6.25}$ | $58.46_{\pm0.79}$ | $19.52_{\pm1.42}$ | $18.66_{\pm1.39}$ | $34.59_{\pm2.07}$ |
| | $\Delta(\uparrow)$ | 2.47 | 2.92 | 7.01 | 7.35 | 4.32 | 1.89 | 3.60 | 2.75 | 3.67 |
| | RAR | $35.59_{\pm2.81}$ | $27.21_{\pm4.36}$ | $46.15_{\pm2.49}$ | $31.97_{\pm5.01}$ | $24.34_{\pm3.73}$ | $36.67_{\pm3.12}$ | $33.35_{\pm3.01}$ | $31.69_{\pm6.47}$ | $41.46_{\pm2.67}$ |
| | w/ Ours | $41.72_{\pm2.59}$ | $32.36_{\pm3.36}$ | $52.11_{\pm5.27}$ | $34.02_{\pm2.84}$ | $28.89_{\pm1.28}$ | $42.71_{\pm2.15}$ | $44.04_{\pm1.14}$ | $39.24_{\pm3.04}$ | $53.46_{\pm3.68}$ |
| | $\Delta(\uparrow)$ | 6.13 | 5.15 | 5.96 | 2.05 | 4.55 | 6.04 | 10.69 | 7.55 | 12.00 |
| Split CIFAR-100 | ER | $9.99_{\pm2.38}$ | $5.22_{\pm1.32}$ | $12.91_{\pm2.51}$ | $9.66_{\pm1.89}$ | $5.27_{\pm1.33}$ | $12.84_{\pm1.74}$ | $9.66_{\pm1.89}$ | $5.21_{\pm1.33}$ | $12.84_{\pm1.74}$ |
| | w/ Ours | $15.33_{\pm5.63}$ | $7.79_{\pm3.69}$ | $16.68_{\pm5.73}$ | $20.92_{\pm3.29}$ | $10.95_{\pm2.10}$ | $20.66_{\pm2.18}$ | $20.90_{\pm3.29}$ | $10.95_{\pm2.10}$ | $20.67_{\pm2.18}$ |
| | $\Delta(\uparrow)$ | 5.34 | 2.57 | 3.77 | 11.26 | 5.68 | 7.82 | 11.24 | 5.74 | 7.83 |
| | DER | $4.83_{\pm0.13}$ | $5.83_{\pm0.13}$ | $7.24_{\pm0.18}$ | $7.83_{\pm0.12}$ | $7.13_{\pm0.13}$ | $7.07_{\pm0.18}$ | $7.83_{\pm0.12}$ | $7.33_{\pm0.12}$ | $9.79_{\pm0.18}$ |
| | w/ Ours | $9.59_{\pm0.43}$ | $7.59_{\pm0.43}$ | $10.34_{\pm0.48}$ | $12.59_{\pm0.43}$ | $9.59_{\pm0.43}$ | $14.44_{\pm0.47}$ | $13.59_{\pm0.33}$ | $12.59_{\pm0.44}$ | $14.34_{\pm0.48}$ |
| | $\Delta(\uparrow)$ | 4.55 | 1.76 | 3.10 | 4.76 | 2.46 | 7.37 | 5.76 | 5.26 | 4.55 |
| | RAR | $4.46_{\pm0.42}$ | $3.34_{\pm0.35}$ | $9.49_{\pm1.72}$ | $8.42_{\pm2.38}$ | $4.82_{\pm1.15}$ | $10.49_{\pm1.48}$ | $7.69_{\pm0.16}$ | $7.97_{\pm0.11}$ | $7.51_{\pm1.05}$ |
| | w/ Ours | $6.01_{\pm0.19}$ | $5.83_{\pm0.17}$ | $11.29_{\pm0.67}$ | $15.54_{\pm4.10}$ | $7.82_{\pm1.78}$ | $15.71_{\pm3.13}$ | $14.27_{\pm0.11}$ | $11.32_{\pm0.12}$ | $13.51_{\pm1.04}$ |
| | $\Delta(\uparrow)$ | 1.55 | 2.49 | 1.80 | 7.12 | 3.00 | 5.22 | 6.58 | 3.35 | 6.00 |
| Split Mini-ImageNet | ER | $2.13_{\pm0.94}$ | $3.11_{\pm0.93}$ | $8.71_{\pm1.11}$ | $2.75_{\pm0.39}$ | $2.63_{\pm0.23}$ | $8.63_{\pm0.66}$ | $2.91_{\pm0.26}$ | $2.76_{\pm0.26}$ | $6.78_{\pm1.56}$ |
| | w/ Ours | $5.83_{\pm0.13}$ | $5.78_{\pm0.12}$ | $10.61_{\pm1.19}$ | $7.01_{\pm0.16}$ | $6.18_{\pm0.16}$ | $11.38_{\pm0.33}$ | $7.21_{\pm0.92}$ | $5.97_{\pm0.86}$ | $11.37_{\pm1.86}$ |
| | $\Delta(\uparrow)$ | 3.70 | 2.67 | 1.90 | 4.26 | 3.55 | 2.75 | 4.30 | 3.21 | 4.59 |
| | DER | $3.42_{\pm1.31}$ | $2.53_{\pm0.11}$ | $4.21_{\pm0.69}$ | $2.02_{\pm0.48}$ | $3.13_{\pm0.46}$ | $4.84_{\pm0.29}$ | $3.77_{\pm0.39}$ | $3.26_{\pm0.37}$ | $4.28_{\pm0.73}$ |
| | w/ Ours | $6.58_{\pm0.71}$ | $5.55_{\pm0.73}$ | $9.73_{\pm0.02}$ | $7.66_{\pm0.77}$ | $6.74_{\pm0.77}$ | $8.19_{\pm0.09}$ | $6.42_{\pm0.18}$ | $6.12_{\pm0.31}$ | $7.05_{\pm0.37}$ |
| | $\Delta(\uparrow)$ | 3.16 | 3.02 | 5.52 | 5.64 | 3.61 | 3.35 | 2.65 | 2.86 | 2.78 |
| | RAR | $3.68_{\pm0.22}$ | $3.48_{\pm0.19}$ | $8.29_{\pm1.02}$ | $2.25_{\pm0.46}$ | $2.99_{\pm0.45}$ | $6.50_{\pm0.86}$ | $3.29_{\pm0.97}$ | $3.15_{\pm0.89}$ | $7.64_{\pm1.01}$ |
| | w/ Ours | $6.82_{\pm0.48}$ | $7.31_{\pm0.11}$ | $11.68_{\pm1.08}$ | $7.67_{\pm0.14}$ | $6.47_{\pm0.13}$ | $9.71_{\pm0.69}$ | $6.97_{\pm1.18}$ | $6.48_{\pm1.17}$ | $10.53_{\pm0.70}$ |
| | $\Delta(\uparrow)$ | 3.14 | 3.83 | 3.39 | 5.42 | 3.48 | 3.21 | 3.68 | 3.33 | 2.89 |

minimum accuracy retained after learning the current task):

$$\text{minAcc}_t = \frac{1}{t-1} \sum_{i=1}^{t-1} \min_{i<j\leq t} \text{Acc}\left(\mathcal{D}_i, f_{\theta_j}\right),$$

$$\text{WCA} = \frac{1}{T} \text{Acc}\left(\mathcal{D}_T, f_{\theta_T}\right) + \left(1 - \frac{1}{T}\right)\text{minAcc}_T . \tag{13}$$

**Evaluation Protocol**. To evaluate the practical performance of each method, we devise a novel evaluation protocol that reflects the impact of *sub-optimal* hyperparameter values. Specifically, we evaluate CL models across different hyperparameter scales, i.e., $\left\{\frac{1}{k}\lambda, \lambda, k\lambda\right\}, k > 1$, where $\lambda$ represents the optimal hyperparameter values obtained from a grid search, and $k$ is a scale factor. By varying the scale of $\lambda$, we are able to examine whether the CL model generalizes well beyond the desired conditions of hyperparameters. In practice, we set $k = 4$ for $\left\{\frac{1}{k}\lambda, \lambda, k\lambda\right\}$, which usually provides a sufficient and meaningful range for examination.

## 5.2 Experimental Results

**Overall Performance**. Table 2 shows the overall performance of *online* CL, where $\lambda$ represents the optimal value of each hyperparameter obtained from a search on the grid. It is evident that the improvement in FAA is consistently significant (e.g., *ER* is improved by $\{5.34\%, 11.26\%, 11.24\%\}$ on Split CIFAR-100), indicating the remarkable benefit of LUNCH plugin through adaptive task weighting. It also shows significant improvements in AAA and WCA (e.g., *RAR* is improved by $\{5.52\%, 3.35\%, 2.78\%\}$ in AAA on Split Mini-ImageNet and by $\{5.15\%, 4.55\%, 7.55\%\}$ in WCA on Split CIFAR-10), suggesting that LUNCH improves the performance of deep models across the training trajectory and in worst-case scenarios. Additionally, each CL method shows considerable variation with different hyperparameters initializations, thus the results from our protocol can be considered as the expected performance of each CL method when applied in real-world scenarios to comprehensively evaluate the generalizability of CL methods.

Additionally, LUNCH significantly improves the performance of the baseline CL methods on different scale benchmarks. Fig. 3 shows the average results and standard deviations of five runs on Split CIFAR-100 and Split ImageNet-R. *RAR w/ Ours* consistently outperforms the original *RAR* in terms of AAA in a series of continual learning tasks. Due to the page limit, additional empirical results in

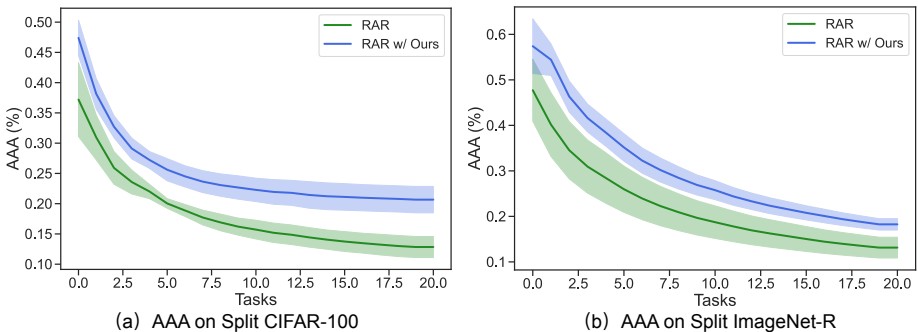

(a) AAA on Split CIFAR-100        (b) AAA on Split ImageNet-R

Figure 3: Performance curves of AAA on benchmark of different scales.

Table 3: Overall performance of *offline* CL with FAA as the evaluation metric. The results of all methods are averaged over five runs with different random seeds and task orders.

| Dataset | ER | ER w/ Ours | $\Delta(\uparrow)$ | DER | DER w/ Ours | $\Delta(\uparrow)$ | RAR | RAR w/ Ours | $\Delta\uparrow$ |
|---|---|---|---|---|---|---|---|---|---|
| Split CIFAR-10 | $65.26_{\pm0.48}$ | $69.49_{\pm0.99}$ | 4.23 | $71.53_{\pm0.49}$ | $74.15_{\pm0.38}$ | 2.62 | $73.52_{\pm0.65}$ | $77.12_{\pm0.59}$ | 3.60 |
| Split CIFAR-100 | $45.14_{\pm0.65}$ | $48.21_{\pm0.73}$ | 3.07 | $57.85_{\pm0.57}$ | $61.07_{\pm0.43}$ | 3.22 | $60.64_{\pm1.01}$ | $63.61_{\pm0.95}$ | 2.97 |

WCA and FAA are provided in Appendix B.1, and these empirical results are consistent with those in AAA.

On the other hand, Table 3 shows the mean and standard deviation of five runs after applying our approach to all baselines (e.g., up to $\sim 4\%$ in FAA) in the *offline* scenarios. From Table 2 and Table 3, we observe that *RAR w/ Ours* often achieves superior performance gains compared to other baselines, mainly because it effectively utilizes advanced data augmentation to capture useful visual information (Zhang et al., 2022). Therefore, we will use RAR as the primary baseline for further analysis in the sequel.

**Ablation Study** We conduct an ablation study to demonstrate the effectiveness of each component, including the uncertain hyperparameters and the temporal ensembling strategy EMA. We compared several variants to analyze their individual contributions. As shown in Table 4, *RAR w/ Unc* introduces optimizable uncertainty into CL-relevant hyperparameters without temporal ensembling, *RAR w/ EMA* only incorporates

Table 4: Ablation study of LUNCH with RAR as baseline.

| Dataset | Metric | RAR | w/ EMA | w/ Unc | w/ Ours |
|---|---|---|---|---|---|
| Split CIFAR-10 | FAA | 42.07 | 46.79 | 49.25 | **55.28** |
| | WCA | 37.41 | 37.81 | 45.81 | **49.97** |
| | AAA | 58.23 | 60.05 | 59.23 | **61.06** |
| Split CIFAR-100 | FAA | 9.19 | 9.43 | 9.33 | **11.77** |
| | WCA | 5.21 | 5.93 | 5.55 | **7.39** |
| | AAA | 11.78 | 12.37 | 12.17 | **14.78** |
| Split Mini-ImageNet | FAA | 3.21 | 4.74 | 4.43 | **7.42** |
| | WCA | 3.26 | 3.32 | 4.69 | **7.12** |
| | AAA | 10.62 | 11.62 | 12.99 | **16.12** |

the temporal ensembling strategy with deterministic hyperparameters, and *RAR w/ Ours* represents the full version of LUNCH that leverages both components. It is obvious that both the learnable uncertain hyperparameters and the temporal ensembling strategy contribute to CL, and removing either results in performance degradation.

Table 5: Impact of sub-optimal hyperparameter values in terms of batch size, learning rate decay and weight decay. We report FAA averaged over five runs with different random seeds and task orders.

| Dataset | ER | ER w/ Ours | $\Delta(\uparrow)$ | DER | DER w/ Ours | $\Delta(\uparrow)$ | RAR | RAR w/ Ours | $\Delta(\uparrow)$ |
|---|---|---|---|---|---|---|---|---|---|
| Split CIFAR-10 | $23.11_{\pm1.21}$ | $27.38_{\pm1.13}$ | 4.27 | $24.11_{\pm1.37}$ | $29.84_{\pm0.84}$ | 5.73 | $33.11_{\pm1.96}$ | $38.49_{\pm2.04}$ | 5.38 |
| Split CIFAR-100 | $8.74_{\pm1.09}$ | $14.99_{\pm1.41}$ | 3.25 | $10.31_{\pm1.11}$ | $13.42_{\pm0.49}$ | 3.11 | $11.47_{\pm0.73}$ | $15.21_{\pm1.12}$ | 3.74 |

Additionally, we evaluate the computational costs of different methods integrating LUNCH (i.e., *w/ Ours*) or not (i.e., *w/o Ours*). As shown in Fig. 4, it is evident that there is minimal difference in time between the *w/o Ours* and *w/ Ours* cases. Our method enhances performance without significantly increasing the extra computational costs, thus making it practical for CL applications with limited computational resources.

**Sensitivity Analysis**. We further extend our evaluation protocol to verify the effectiveness of LUNCH in improving the robustness of other hyperparameters, such as the learning rate decay, batch size, and weight decay. These hyperparameters are also relevant to CL yet in an implicit manner (Mirzadeh et al., 2020; Cha & Cho, 2024). For ease of implementation, we randomly sample a set of sub-optimal hyperparameters from predefined values (i.e., moderately rescaling optimal hyperparameters in Appendix B.2), and average multiple runs to evaluate the practical performance of each method, thus avoiding overestimation of CL capabilities in practical use. As shown in Table 5, we observe that our approach achieves better performance compared to

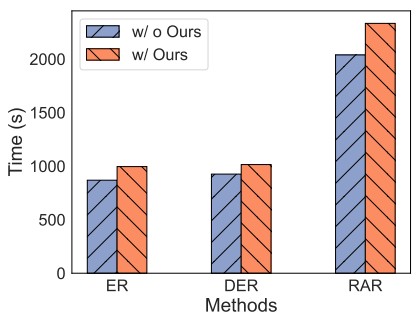

Figure 4: Comparison of extra computational time with and without LUNCH (Ours).

the corresponding baselines, and in particular the improvement of RAR is more significant. Aligning with the results of previous analysis, LUNCH can significantly alleviate hyperparameter sensitivity, especially with more advanced methods.

Of note, the above results pose the particular challenge of hyperparameter sensitivity in CL. A majority of CL methods are developed with the "optimal" hyperparameter values obtained from a grid search, and thus may fail to produce desirable results based on such hyperparameter values when there exist large changes in data distribution. In contrast, our protocol of evaluating hyperparameter sensitivity can faithfully reflect this issue, and the proposed LUNCH can effectively address it as a plug-in manner.

## 6 DISCUSSION AND CONCLUSION

In this work, we propose an innovative approach for adaptive weighting of task contributions in CL, which optimizes CL-relevant hyperparameters according to training progress and thus alleviates catastrophic forgetting. In particular, we formulate each CL-relevant hyperparameter as a function of learnable uncertainty under homoscedastic assumption, and ensure training stability via the exponential moving average of network parameters along the training trajectory. Extensive empirical results demonstrate the benefits of our approach in improving the effectiveness and robustness of the corresponding baselines in a variety of CL scenarios. We hope that this work will inspire more explorations of adaptive weighting in CL scenarios and shed light on building general, practical, and effective CL methods in this field.

This work also has some potential limitations. First, we follow the implementations of representative online CL methods, which mainly employ ResNet-based backbone. We leave examining the effectiveness of uncertain hyperparameters with ViT-based backbone as a potential future work. Second, we focus on CL methods within the same parameter space. Therefore, the proposed uncertain hyperparameters are not applicable to many architecture-based CL methods that focus on multiple parameter spaces. Since this work is essentially a fundamental research in machine learning, the potential negative impact is not obvious at the current stage.

## 7 REPRODUCIBILITY

The source code of our experiments is included in the supplymentary materials. The theoretical results, including key assumptions and proofs, are provided in Appendix A. The specific hyperparameters used in our experiments can be found in Section 5.1 and the associated yml configuration files are provided in the source code. These resources allow for a comprehensive replication of our results.

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

## A  UNCERTAINTY DERIVATION

### A.1  REGRESSION TASKS

The derivation of uncertainty in regression is shown in followings.

$$p\left(y \mid f_\theta(x), \sigma^2\right) = \mathrm{Lap}\left(f_\theta(x), \sigma^2\right),$$

$$= \frac{1}{2\sigma^2} \exp\left(-\frac{\|y - f_\theta(x)\|^2}{\sigma^2}\right)$$

$$\log p\left(y \mid f_\theta(x), \sigma^2\right) = -\frac{1}{\sigma^2}|y - f_\theta(x)| - \log 2\sigma^2, \tag{14}$$

$$-\log p\left(y \mid f_\theta(x), \sigma^2\right) = \frac{1}{\sigma^2}|y - f_\theta(x)| + \log 2\sigma^2,$$

$$\propto \frac{1}{\sigma^2}|y - f_\theta(x)| + \log \sigma^2.$$

Thus, we can derive the joint distribution of multiple regression tasks:

$$p\left(y_1, y_2, \dots \mid f_\theta(x), \sigma_1^2, \sigma_2^2, \dots\right) = p\left(y_1 \mid f_\theta(x), \sigma_1^2\right) \cdot p\left(y_2 \mid f_\theta(x), \sigma_2^2\right) \cdot \dots$$
$$= \mathrm{Lap}\left(y_1; f_\theta(x), \sigma_1^2\right) \cdot \mathrm{Lap}\left(y_2; f_\theta(x), \sigma_2^2\right) \cdot \dots \tag{15}$$

$$-\log p\left(y_1, y_2, \dots \mid f_\theta(x), \sigma_1^2, \sigma_2^2 \dots\right) \propto \frac{1}{\sigma_1^2}\|y_1 - f_\theta^1(x)\|^2 + \frac{1}{\sigma_2}\|y_2 - f_\theta^2(x)\|^2$$
$$+ \log \sigma_1^2 + \log \sigma_2^2 + \dots \tag{16}$$

### A.2  CLASSIFICATION TASKS

We use the maximum likelihood estimation as our objective function:

$$p\left(y = c \mid f_\theta(x), \sigma^2\right) = \frac{\exp\left(f_\theta^c(x)/\sigma^2\right)}{\sum_{c'} \exp\left(f_\theta^{c'}(x)/\sigma^2\right)},$$

$$\log p\left(y = c \mid f_\theta(x), \sigma^2\right) = \frac{1}{\sigma^2}f_\theta^c(x) - \log \sum_{c'} \exp\left(\frac{1}{\sigma^2}f_\theta^{c'}(x)\right). \tag{17}$$

For classification problems, we use the cross entropy loss function:

$$p(y = c \mid x, \theta) = -\log\left(\mathrm{Softmax}\left(y = c, f_\theta(x)\right)\right),$$

$$\log p(y = c \mid x\theta) = \log\left(\sum_{c'} \exp(f_\theta^{c'}(x))\right) - f_\theta^c(x). \tag{18}$$

We can rewrite and derive the corresponding forms:

$$\log p\left(y = c \mid f_\theta(x), \sigma^2\right) = \frac{1}{\sigma^2}f_\theta^c(x) - \log \sum_{c'} \exp\left(\frac{1}{\sigma^2}f_\theta^{c'}(x)\right)$$

$$+ \frac{1}{\sigma^2}\log \sum_{c'} \exp\left(f_\theta(x)\right) - \frac{1}{\sigma^2}\log \sum_{c'} \exp\left(f_\theta(x)\right)$$

$$= \frac{1}{\sigma^2}f^{c_\theta}(x) - \frac{1}{\sigma^2}\log \sum_{c'} \exp\left(f_\theta(x)\right) \tag{19}$$

$$+ \log\left(\sum_{c'} \exp\left(f_\theta(x)\right)\right)^{\frac{1}{\sigma^2}} - \log \sum_{c'} \exp\left(\frac{1}{\sigma^2}f_\theta^{c'}(x)\right)$$

$$= -\frac{1}{\sigma^2}\mathcal{L}(y = c, \theta) - \log \frac{\sum_{c'} \exp\left(\frac{1}{\sigma^2}f_\theta^c(x)\right)}{\left(\sum_{c'} \exp\left(f_\theta'(x)\right)\right)^{\frac{1}{\sigma^2}}}.$$

Under the assumption $\left( \sum_{c'} \exp \left( f_\theta^{c'}(x) \right) \right)^{\frac{1}{\sigma^2}} \approx \frac{1}{\sigma^2} \sum_{c'} \exp \left( \frac{1}{\sigma^2} f_\theta^{c'}(x) \right)$ that allows us to simplify the objective function:

$$- \log p \left( y = c \mid f_\theta(x), \sigma^2 \right) \approx \frac{1}{\sigma^2} p(y = c \mid x, \theta) + \log \sigma^2. \tag{20}$$

Thus, we can derive the joint distribution of multiple classification tasks (take two tasks as examples):

$$p \left( \theta, \sigma_1^2, \sigma_2^2 \right) \propto \frac{1}{\sigma_1^2} \mathcal{L}_1(\theta) + \frac{1}{\sigma_2^2} \mathcal{L}_2(\theta) + \log \sigma_1^2 + \log \sigma_2^2 + \ldots, \tag{21}$$

$$p \left( y_1, y_2 \mid f_\theta(x), \sigma_1^2, \sigma_2^2 \right) = \text{Softmax} \left( y_1; \frac{1}{\sigma_1^2} f_\theta(x) \right) \cdot \text{Softmax} \left( y_2; \frac{1}{\sigma_2^2} f_\theta(x) \right), \tag{22}$$

$$- \log p \left( y_1, y_2 \mid f_\theta(x), \sigma_1^2, \sigma_2^2 \right) = \log \text{Softmax} \left( y_1; \frac{1}{\sigma_1^2} f_\theta(x) \right) + \log \text{Softmax} \left( y_2; \frac{1}{\sigma_2^2} f_\theta(x) \right) \tag{23}$$

$$- \log p \left( y_1, y_2 \ldots \mid f_\theta(x), \sigma_1^2, \sigma_2^2 \ldots \right) = \frac{1}{\sigma_1^2} \mathcal{L}_1(\theta) + \frac{1}{\sigma_2^2} \mathcal{L}_2(\theta) + \frac{1}{2} \log \sigma_1^2 + \frac{1}{2} \log \sigma_2^2 + \ldots. \tag{24}$$

## B    MORE EXPERIMENTAL DETAILS

### B.1    MORE EXPERIMENTAL RESULTS ABOUT PERFORMANCE CURVES

Fig. 5 shows the average results and standard deviations of five runs in WCA and AAA on Split CIFAR-100 and Split ImageNet-R. *RAR w/ Ours* achieves superior performance in both WCA and FAA compared to *RAR*, aligning with the main results of AAA in Fig. 3.

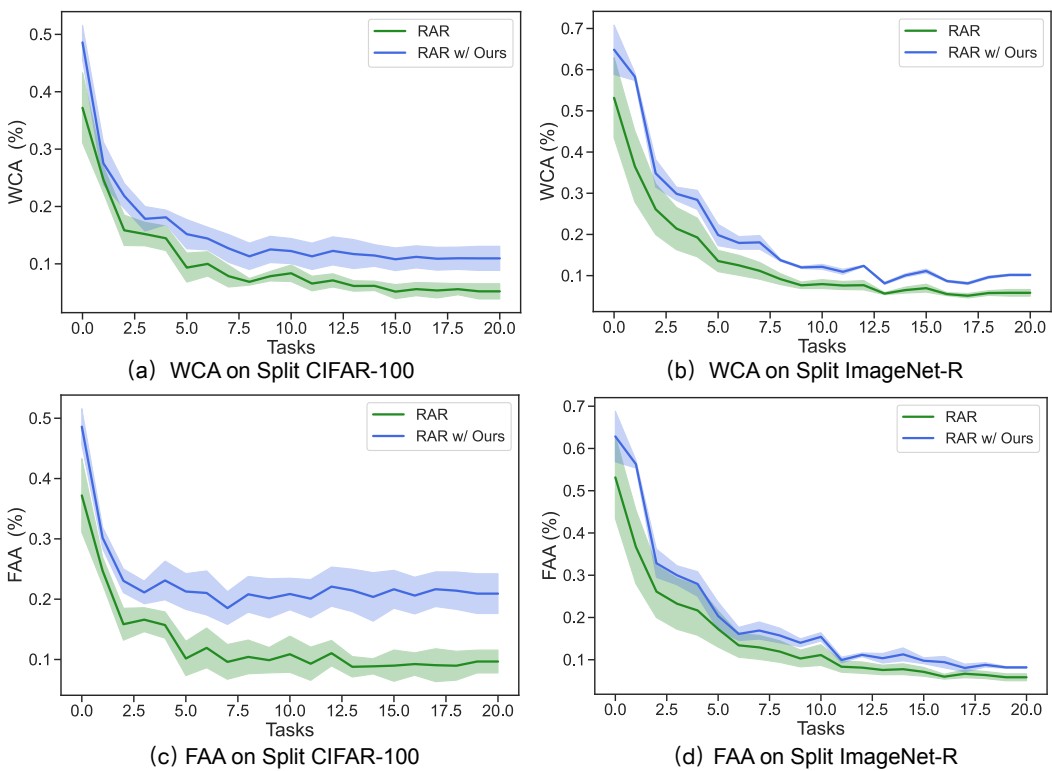

Figure 5: The performance curves of WCA and FAA for RAR with LUNCH plugin.

## B.2 HYPERPARAMETER SETS

All hyperparameters for the sensitivity analysis in our proposed protocol were sampled from the hyperparameter set in Table 6.

Table 6: The pre-defined set of hyperparameter values.

| Hyperparameters | Value sets |
|---|---|
| Learning rate decay | [0.025, 0.05, 0.1, 0.2, 0.4] |
| Batch size | [32, 64, 128, 256, 512] |
| Weight decay | [0.0005, 0.001, 0.002, 0.004] |

