# OpenReview forum: "LUNCH: Adaptive Balancing of Continual Learning via Hyperparameter Uncertainty"
_ICLR.cc/2025/Conference — ICLR 2025 Conference Withdrawn Submission_

### Official Review · Reviewer_Bh6X · 2024-11-02

**Soundness:** 2
**Presentation:** 2
**Contribution:** 2
**Rating:** 3
**Confidence:** 5

**Summary:**

Continual learning (CL) aims to sequentially learn tasks while preventing catastrophic forgetting of previous ones. The proposed approach, Learning UNCertain Hyperparameters (LUNCH), introduces adaptive balancing by treating CL hyperparameters as functions of optimizable uncertainty, enhancing robustness in online learning scenarios. Extensive experiments demonstrate LUNCH's effectiveness, showing significant performance improvements in both online and offline CL settings.

**Strengths:**

**[Adaptive Strategy]** This method helps to improve the performance of regularization-based CL methods by searching the optimal hyper-parameters, and shows the large improvement in Tab.2.

**[Various Experiments]** To validate the effectiveness of proposed method, there were various experiments. I was impressed that adequate hyperparameters can improve the CL method performance with large gap.

**Weaknesses:**

**[Similar Work]** I saw the paper that is adjusting the EMA parameter in online CL. The paper is not focused on the searching the best hyper-parameters, whereas the authors stated that EMA parameter is crucial for the model performance. In this context (updating ema parameter), the methodology of LUCNH paper is similar to previous paper.

[1] ONLINE BOUNDARY-FREE CONTINUAL LEARNING BY SCHEDULED DATA PRIOR, ICLR 2023.

**[Narrow Scope]** To the best of my understanding, this method can be applicable to the regularization-based CL methods. However, there are many different approaches for CL, such as L2P [2], DualPrompt [3], DAP [4], etc. I think only applying the proposed method into regularization-based CL seems to be too narrow contribution.

[2] Learning to Prompt for Continual Learning, CVPR 2022

[3] DualPrompt: Complementary Prompting for Rehearsal-free Continual Learning, ECCV 2022

[4] Generating instance-level prompts for rehearsal-free continual learning, ICCV 2023

**Questions:**

This part is for the minor comments that are not directly related to the contents of this paper.

**[Orphans]** There are several orphans in this paper. It should be modified for paper completeness. Please refer this site. https://en.wikipedia.org/wiki/Widows_and_orphans

**[Reference position]** Currently, Reference section is in the last of the page, but it should be started with the new page.

**[Appendix Autoref]** Please use \autoref function when referring appendix. Currently, there is no functionality to go to the appendix section automatically.

**[Space]** There are several redundant spaces, such as eq. 11 and 12. I think it can be merged to save the space of main paper.

---

### Official Review · Reviewer_bqvy · 2024-11-02

**Soundness:** 2
**Presentation:** 3
**Contribution:** 2
**Rating:** 5
**Confidence:** 3

**Summary:**

The paper developed the Learning UNCertain Hyperparameters (LUNCH) algorithm to address catastrophic forgetting in CL. The idea is to dynamically adjusts task balancing by treating hyperparameters as functions of learnable uncertainty. As a result, hyperparameters can be adapted based on uncertainty, enhancing task retention flexibility to mitigate forgetting effect. In order to make LUNCH stable, an exponential moving average of network parameters are used and a sensitivity analysis protocol is offered to fine-tune hyperparameters effectively. Experimental results on three common CL benchmark show that LUNCH is effective and leads to improved performance

**Strengths:**

1. The paper reads well and is straightforward to follow.

2. Experiments on three common benchmarks are reported.

3. The code is provided and although I did not run it, it looks to be easy to run which makes reproducing the results straightforward.

**Weaknesses:**

1. Motivation is unclear and it is not clear what challenge this work addresses that existing methods do not address. Since CL is an established field with so many existing work, new works should distinguish themselves by addressing an unaddressed challenge or demonstrate significant performance boost.

2. I am concerned that basslines for comparison are limited and the used baselines are outdated given the fast and dynamic progress in CL. Works with SOTA performance from 2023-2024 should be used in comparisons. A quick search in the literature demonstrates that there are works with better performance compared to the baselines used in the paper.

3. Analytic experiments do give much insight why the proposed method is effective? Under what circumstances it can work and what are the limitations. For example, what are the criteria under which LUNCH might fail to work? What are some settings in which it can outperform existing approaches with a wider margin?

**Questions:**

1. A major concern that I have is that CL is an extremely crowded field with many existing CL methods that report their performance on three benchmarks used in the experiments. What is the challenge that LUNCH is addressing that existing methods do not address? What is the addition to the field?

2. What is the logic for selecting the baselines that were used for comparison? There are hundreds of existing works that report their performance on the used benchmarks in the experiments. It is OK to include some old works in comparison to reflect the progress in the field since the introduction of pioneer works, but it is essential to include recent methods with SOTA performance to demonstrate how competitive the proposed approach is. Even is other methods might be very different from LUNCH, they still be used for comparison.

3. Since forgetting is a major focus of the paper, I was wondering why metrics such as backward transfer have not been reported to offer evaluations targeted on measuring forgetting?

4. Can this method be used with larger transformer-based models? Currently, the interest in CL has shifted to these models.

5. I think MTL and sequential task learning performance need to be reported to have the upperbound and the lowerbound performance as well.

---

> ### Comment · Reviewer_bqvy · 2024-11-26
> **Post-Rebuttal Rating**
>
> No response has been submitted and hence, I maintain my initial score.

---

### Official Review · Reviewer_Yv3P · 2024-11-04

**Soundness:** 2
**Presentation:** 3
**Contribution:** 2
**Rating:** 3
**Confidence:** 4

**Summary:**

This paper proposes Learnable Uncertain Hyperparameters (LUNCH), a method that replaces fixed hyperparameters with learnable parameters for continual learning (CL) algorithms. In LUNCH, the hyperparameters associated with three loss terms defined in a unified CL framework—such as for new tasks, output space, and parameter space—are formulated as optimizable parameters based on homoscedastic aleatoric uncertainty, allowing them to be adjusted during the training process for each task. Additionally, whenever a new task is introduced, the uncertainty parameters are reinitialized to enable the model to learn unique hyperparameters for each task. To address the "stability gap" that may arise during this process, they apply an Exponential Moving Average (EMA) technique. Extensive experiments in both online and offline CL scenarios demonstrate that LUNCH consistently enhances overall performance.

**Strengths:**

1. This paper addresses a critical yet often overlooked issue in continual learning research: the challenge of setting hyperparameters for continual learning algorithms. It introduces a novel approach, called LUNCH, designed to learn these hyperparameters effectively.

2. The proposed LUNCH algorithm leverages a unified CL framework and homoscedastic aleatoric uncertainty to define the hyperparameters of each loss term as learnable parameters, providing a new method for optimizing these parameters. This approach offers a fresh alternative to traditional hyperparameter tuning and selection methods.

3. Applying LUNCH to both offline and online continual learning settings has shown consistent performance improvements across diverse experiments. Additionally, through ablation studies, cost comparisons, and sensitivity analyses, we demonstrate the effectiveness of LUNCH across multiple dimensions.

**Weaknesses:**

1. I believe the main weakness of this paper lies in its focus solely on hyperparameters for three loss terms, while overlooking "basic" hyperparameters (e.g., mini-batch size, learning rate, and optimizer) that are typically set differently in each CL algorithm to achieve best results. In fact, these basic hyperparameters are known to significantly influence overall performance, comparable to the impact of hyperparameters specific to CL algorithms, as discussed in (Cha & Cho, 2024). Therefore, for LUNCH to serve as a comprehensive solution for hyperparameter tuning and setting in CL, it is essential to consider and discuss how these basic hyperparameters should be tuned by the proposed method.

2. The algorithms and experimental scenarios considered for applying LUNCH are insufficient. Although LUNCH is not designed specifically for a particular CL algorithm, without theoretical proof of its universality, further experiments are needed to demonstrate that the proposed approach can be applied to various CL algorithms and scenarios. For instance, in online CL, additional algorithms such as EARL, REMIND, SCR, and X-DER considered in [1] should be considered. For offline CL, algorithms like BiC, PODNet, iCaRL, and WA from [2] should be examined. Furthermore, it is known that each algorithm’s performance can vary depending on the number of tasks or the dataset used (See [2] and (Cha & Cho, 2024)). Thus, additional experiments on varying numbers of tasks and datasets like ImageNet (or ImageNet-100) are essential to verify if LUNCH can be broadly applied across diverse CL scenarios.

3. Based on Line 118, this paper appears to focus on class-incremental learning. However, some of the results in Table 2 seem unusual:
   - 3.1) The FAA scores for ER and DER are much lower than expected (e.g., split CIFAR-10) compared to Table 1 of [1].
   - 3.2) Although ($\lambda$) is the optimal hyperparameter, in many cases, using \(1/4 $\times$ $\lambda$\) yields better performance, even without LUNCH.
   - 3.3) Furthermore, contrary to [1] and commonly known findings, ER shows superior performance compared to other algorithms even without LUNCH.

4. In the ablation study results in Table 4, it appears that the performance improvement from EMA often exceeds that of Unc in many cases. To clarify the effectiveness of Unc, I suggest conducting experiments where EMA is applied only at specific iterations after the start of each task’s training (e.g., applying EMA only at \(1/4 $\times$ $I$\), where \($I$\) is the total training iteration for task \($t$\), when stability gap is most likely to occur). I believe this additional ablation study would help better demonstrate the advantages of the Unc.


[1] Learning Equi-angular Representations for Online Continual Learning, CVPR 2024.

[2] PyCIL: A Python Toolbox for Class-Incremental Learning, Arxiv.

**Questions:**

I wrote all weaknesses and questions in the Weakness section; please refer to it. Overall, I commend the authors for highlighting the critical issue of hyperparameter tuning in continual learning research and for proposing an algorithm to address it. However, I find three main weaknesses: (1) the proposed algorithm only considers the hyperparameters of three loss terms, (2) the range of CL algorithms and scenarios considered is limited, and (3) some experimental results differ from known performance. I look forward to any clarifications regarding potential misunderstandings on my part and to the author's response.

---

### Official Review · Reviewer_Gnbq · 2024-11-04

**Soundness:** 2
**Presentation:** 3
**Contribution:** 2
**Rating:** 3
**Confidence:** 3

**Summary:**

The paper introduces a novel approach to continual learning (CL) that adaptively balances task contributions by optimizing hyperparameters based on model uncertainty.  The method incorporates an exponential moving average to stabilize training and reduce catastrophic forgetting. Extensive experiments of LUNCH across standard benchmarks significantly improves accuracy and robustness in CL settings.

**Strengths:**

1. LUNCH introduces a novel approach by adaptively adjusting hyperparameters based on uncertainty, which brings a fresh perspective to continual learning.
2. The paper is well-organized and clearly articulates this approach.
3. LUNCH demonstrates consistent improvements across a range of metrics showcasing its robustness in varied CL conditions.
4. The method easily integrates well with existing CL techniques, such as Empirical Replay (ER) and Dark Experience Replay (DER), enhancing these methods.
5. The paper provides a thorough evaluation with sensitivity analyses on suboptimal hyperparameter values, revealing LUNCH’s stability and robustness.

**Weaknesses:**

- The paper does not examine LUNCH’s robustness under significant shifts in data distribution, a common occurrence in real-world continual learning environments. Without tests for robustness to such shifts, it remains unclear how well LUNCH maintains performance when faced with sudden changes in data characteristics. For example, the authors could potentially test the method with gradual concept drift, sudden concept shift, or domain adaptation scenarios between tasks to simulate real-world data variability.

-  LUNCH’s performance is tested on relatively short task sequences. The authors did not discuss the computational feasibility and potential challenges of scaling LUNCH to much longer sequences (e.g., 50+ or 100 tasks). To strengthen the paper I request authors to evaluate LUNCH across an extended sequence of 50+ or 100 tasks to gauge long-term stability, retention, and resistance to catastrophic forgetting.

-  It will be interesting to see how LUNCH performs in scenarios where imbalance in task frequencies ( some tasks appearing more frequently than others) is prevalent. For example, the authors can consider evaluating LUNCH on a sequence where 20% of tasks appear 80% of the time, mirroring real-world power law distributions.

**Questions:**

-  The effectiveness of individual uncertainty components is unclear. It will be interesting to conduct an ablation study isolating individual uncertainty components (e.g., output-space vs. parameter-space uncertainty) to assess their distinct impact on adaptability. This would clarify which elements drive LUNCH’s performance most significantly.

---

### Note · Authors · 2024-12-13

I have read and agree with the venue's withdrawal policy on behalf of myself and my co-authors.